# Kidney Involvement in Hypocomplementemic Urticarial Vasculitis Syndrome—A Case-Based Review

**DOI:** 10.3390/jcm9072131

**Published:** 2020-07-06

**Authors:** Oana Ion, Bogdan Obrișcă, Gener Ismail, Bogdan Sorohan, Sonia Bălănică, Gabriel Mircescu, Ioanel Sinescu

**Affiliations:** 1Department of Nephrology, Fundeni Clinical Institute, 022328 Bucharest, Romania; oana.catalina_ion@yahoo.ro (O.I.); obriscabogdan@yahoo.com (B.O.); bogdan.sorohan@yahoo.com (B.S.); sonia.balanica@gmail.com (S.B.); 2Department of Uronephrology, “Carol Davila” University of Medicine and Pharmacy, 020021 Bucharest, Romania; gmircescu@hotmail.com (G.M.); umfisinescu@gmail.com (I.S.); 3“Dr. Carol Davila” Teaching Hospital of Nephrology, 010731 Bucharest, Romania; 4Center of Uronephrology and Renal Transplantation, Fundeni Clinical Institute, 022328 Bucharest, Romania

**Keywords:** hypocomplementemic urticarial vasculitis syndrome (HUVS), kidney involvement, crescentic glomerulonephritis, nephrotic syndrome, anti-C1q antibodies

## Abstract

Hypocomplementemic urticarial vasculitis syndrome (HUVS), or McDuffie syndrome, is a rare small vessel vasculitis associated with urticaria, hypocomplementemia and positivity of anti-C1q antibodies. In rare cases, HUVS can manifest as an immune-complex mediated glomerulonephritis with a membranoproliferative pattern of injury. Due to the rarity of this disorder, little is known about the clinical manifestation, pathogenesis, treatment response and outcome of such patients. We describe here three cases of HUVS with severe renal involvement. These patients had a rapidly progressive form of glomerulonephritis with severe nephrotic syndrome against a background of a membranoproliferative pattern of glomerular injury with extensive crescent formation. Therefore, these patients required aggressive induction and maintenance immunosuppressive therapy, with a clinical and renal response in two patients, while the third patient progressed to end-stage renal disease. Because of the rarity of this condition, there are few data regarding the clinical presentation, pathology and outcome of such patients. Accordingly, we provide an extensive literature review of cases reported from 1976 until 2020 and place them in the context of the current knowledge of HUVS pathogenesis. We identified 60 patients with HUVS and renal involvement that had adequate clinical data reported, out of which 52 patients underwent a percutaneous kidney biopsy. The most frequent renal manifestation was hematuria associated with proteinuria (70% of patients), while one third had abnormal kidney function on presentation (estimated glomerular filtration (GFR) below 60 mL/min/1.73 m^2^). The most frequent glomerular pattern of injury was membranoproliferative (35%), followed by mesangioproliferative (21%) and membranous (19%). Similar to other systemic vasculitis, renal involvement carries a poorer prognosis, but the outcome can be improved by aggressive immunosuppressive treatment.

## 1. Introduction

Hypocomplementemic urticarial vasculitis syndrome (HUVS) is a rare autoimmune hypocomplementemic small vessels vasculitis, affecting skin, joints, eyes, lungs and kidneys, first described by McDuffie et al., (1973) [1,2,3,4]. Thereafter, a high titer of anti-C1q antibodies was described and the diagnostic criteria were proposed by Schwartz et al., (1982) [1]. More recently, HUVS was defined based on histopathology as “anti-C1q (leukocytoclastic) vasculitis” and classified as “small vessels vasculitis” accompanied by urticaria, hypocomplementemia and anti-C1q antibodies [5].

Although anti-C1q antibodies and complement consumption are clearly involved, HUVS pathogenesis is unclear and what triggers these abnormalities is a matter of debate [6,7].

Probably because HUVS is a rare condition and kidney biopsy was not consistently performed, kidney involvement was reported with variable frequency (14–50%), the histopathology and clinical pattern as well as outcome of kidney involvement are unclear [2,3,8]. Various schedules of immunosuppression were tried with mixed results.

Here, we report three cases of hypocomplementemic urticarial vasculitis syndrome with renal involvement; we undertook a literature search (1976–2020), focusing on cases for which individual data on renal involvement were available, aiming to better profile the histopathology and clinical presentation as well as the outcome of kidney involvement in HUVS.

## 2. Case Presentations

### 2.1. Case 1

A 47-year-old woman was admitted to our department with left lumbar pain and macroscopic hematuria. Her medical history was unremarkable. Physical examination revealed pale skin, without peripheral edema, a blood pressure of 130/80 mmHg and a diuresis of 1000 mL/day.

The initial evaluation showed increased serum creatinine (4.29 mg/dL) and nephrotic syndrome (proteinuria 13.4 g/day and serum albumin 2.8 g/dL). Urinalysis showed hematuria with dysmorphic red blood cells and acanthocytes. At ultrasound examination, the kidneys appeared normal. The patient had inflammatory syndrome (C-reactive protein, 37.5 mg/L) and moderate, normocytic anemia.

Antinuclear antibodies (ANA), anti-ds-DNA (double-stranded DNA), anti-Ro, anti-La, anti-Sm, anti-RNP, anti-β2-glycoprotein 1, antiphospholipid, cANCA, pANCA, anti-GBM antibodies, cryoglobulins and rheumatoid factor were not detected. Tests for hepatitis viruses B (HBV), C (HCV) and HIV were negative and serum protein electrophoresis did not reveal a monoclonal spike. The titer of anti-C1q antibodies was high (75.8 IU/mL), while serum complement fractions were markedly decreased (C3—58.4 mg/dL; C4—9.24 mg/dL) (Table 1).

A kidney biopsy was performed. By light microscopy (LM), and the pattern was membrano-proliferative with diffuse lobular accentuation (mesangial and endocapillary hypercellularity). Additionally, fibrinoid necrosis, capillary wall rupture and extra-capillary hypercellularity were seen in over 50% of the examined glomeruli. There was also moderate tubular atrophy and interstitial fibrosis. In immunofluorescence (IF), the staining was moderate for IgG, IgM, C1q and C3, in the mesangium and along the glomerular capillary walls. By electron microscopy (EM), mesangial and subendothelial electron-dense deposits were identified in addition to severe endothelial injury. There was also severe podocyte injury with diffuse foot process effacement (Figure 1).

Thus, the clinical and pathological features were consistent with an immune complex mediated membrano-proliferative glomerulonephritis with positive C1q immunostaining, crescent formation, circulating anti-C1q antibodies and complement activation. Because anti-ds-DNA antibodies were absent, a lupus nephritis was excluded and, as serum complement was low and the glomerular lesions were intensively exudative, a C1q glomerulopathy was also excluded [9]. A membrano-proliferative pattern associated with complement activation by anti-C1q antibodies as descried by Strife et al. could be a possibility because skin lesions and other organs’ involvement were absent, but cases in which kidney lesions preceded the other HUVS manifestations were reported [10,11].

Aggressive immunosuppressive (IS) induction therapy with monthly intravenous (IV) pulses—Cyclophosphamide 15 mg/kg/month and Methylprednisolone 500 mg/day for 3 days, followed by oral Prednisolone (initially 64 mg/day) was started. However, after the initial course of therapy, renal function worsened (serum creatinine rose to 6.9 mg/dL). Accordingly, plasma exchange (five sessions) was initiated, resulting in an important improvement of renal function: serum creatinine and anti-C1q antibody titer decreased to 3.2 mg/dL and 3 IU/mL. Thereafter, renal function continued to improve.

After six months, a partial clinical remission was achieved: creatinine level returned to normal, nephrotic syndrome was partially remitted, while serum C3 and C4 normalized. Maintenance therapy (Azathioprine 100 mg/day and oral Prednisolone 16 mg/day) was started.

At last follow-up 27 months after diagnosis, the patient had normal renal function, proteinuria was 1.8 g/day, anti-C1q antibodies were absent, and serum complement fractions were normal (Table 1).

### 2.2. Case 2

An active smoker 47-year-old woman with a history of type 1 diabetes mellitus was referred to our department for evaluation of a new-onset arterial hypertension and peripheral edema. At physical examination, she had important lower limbs and palpebral edema, a discreet erythematous wheal-like eruption on the face, associated with angioedema. Blood pressure was 140/80 mmHg and she had a diuresis of 2000 mL/day. At ultrasound examination, the kidneys appeared normal.

Laboratory tests showed mild renal dysfunction (serum creatinine level of 1.27 mg/dL) and full nephrotic syndrome (serum albumin 1.3 g/dL and proteinuria 6.4 g/day). Urinalysis revealed microscopic hematuria, dysmorphic red blood cells and leukocyturia. The patient had mild inflammatory syndrome and normocytic, normochromic anemia.

Immunological work-up—ANA, anti-ds-DNA antibodies, anti-Ro antibodies, anti-La antibodies, pANCA, cANCA, anti-GBM antibodies, anti-PLA2R antibodies, rheumatoid factor and cryoglobulins—was negative, except for a high anti-C1q antibodies titer (356 IU/mL). Complement fractions were low (C3 41 mg/dL; C4 8.94 mg/dL). Tests for HBV, HCV and HIV were negative (Table 1).

As microscopic hematuria with new-onset nephrotic syndrome and high titer of anti-C1q antibodies with complement consumption made diabetic nephropathy unlikely, a kidney biopsy was performed.

LM identified a membranoproliferative pattern—accentuation of glomerular lobules, and extensive endocapillary and mesangial hypercellularity with occlusion of some capillary loops—accompanied by vasculitis features, e.g., fibrinoid necrosis, capillary wall rupture and cellular crescents. However, lesions suggestive of diabetic nephropathy—mesangial matrix expansion, tubular basement membrane and Bowman capsule thickening, arteriolar hyalinosis and arteriosclerosis—were also seen and graded as class II diabetic nephropathy. By IF, a moderate, granular staining for IgG, IgM, C1q and C3 in the mesangium and along the glomerular capillary walls was identified, consistent with an immune-complex mediated glomerulonephritis. By EM, mesangial and subendothelial electron-dense deposits were identified in addition to severe podocyte injury with diffuse foot process effacement and diffuse glomerular basement membrane thickening (Figure 2).

Accordingly, the patient had an immune-complex mediated membranoproliferative glomerulonephritis with crescent formation, superimposed on a class II diabetic nephropathy.

The diagnosis of HUVS was confirmed, as Schwartz criteria, two major (urticarial exanthema, hypocomplementemia) and two minor (glomerulonephritis and a high anti-C1q antibody titer), were fulfilled [1].

Aggressive induction IS therapy—monthly IV pulses of Cyclophosphamide (15 mg/kg/month, cumulative dose of 6 g) and a short course of IV Methylprednisolone (stopped due to difficult glycemic control)—was started. Subsequently, the patient received maintenance therapy with Rituximab (100 mg/every three months) with partial renal response.

At last follow-up 16 months after initiation of IS therapy, renal function recovered (serum a creatinine 0.85 mg/dL) and the nephrotic syndrome was in partial remission (proteinuria 2.6 g/day and serum albumin 3.3 g/dL) (Table 1).

### 2.3. Case 3

A 47-year-old woman was admitted to the Nephrology Department with generalized pitting edema, oliguria and fatigue. Her medical history included urticarial-like flares, with a recurrent pattern in the last four years, without an identifiable cause. Uterine fibroids and persistent anemia were diagnosed five years ago.

Physical examination revealed pale skin, with urticarial–like flare involving the face and neck (Figure 3). The wheals were nonpruritic, mildly erythematous and palpable, and were associated with a discreet angioedema of the lips (Figure 3). On admission, the blood pressure was 140/80 mm Hg and breath sounds were absent at lung bases. At ultrasound examination, the kidneys appeared normal.

Kidney function was altered (serum creatinine 5.96 mg/dL). Notably, three months prior to admission, serum creatinine was normal and had started to rise in the last two months. Proteinuria was 4 g/day and serum albumin 2.8 g/dL. The urinary sediment was active: microscopic hematuria with dysmorphic red blood cells and acanthocytes, red blood cells casts and leukocyturia (Table 1).

Complete blood cell count revealed normocytic, normochromic anemia, mild thrombocytopenia (130,000 platelets/µL) and leukocytosis with neutrophilia (13,660 WBC/µL, 70% neutrophils). Inflammation was moderate.

Immunological work-up for antinuclear antibodies, anti-ds-DNA antibodies, anti-Ro antibodies, anti-La antibodies, anti-GBM antibodies, cryoglobulins and rheumatoid factor was negative. There was a marked consumption of serum complement fractions C4 (3.6 mg/dL) and C3 (18.7 mg/dL), with a high titer of anti-C1q antibodies (112.8 U/mL). Tests for HBV, HCV and HIV were negative.

The diagnosis of HUVS was confirmed, as Schwartz criteria, two major (recurrent urticarial flares chronic urticarial exanthema, hypocomplementemia) and two minor (glomerulonephritis and a high anti-C1q antibody titer), were fulfilled [1].

Induction therapy—IV pulses of Methylprednisolone and Cyclophosphamide—was started but the patient developed pancytopenia with severe anemia (Hb 5.4 g/dL) and thrombocytopenia (23,000/µL) which precluded renal biopsy. Oliguria persisted, and the renal function did not improve.

One month after admission, hemodialysis was initiated and, at last follow-up, the renal function had not recovered. The patient is currently on chronic hemodialysis (Table 1).

## 3. Discussion

### 3.1. HUVS Clinical Presentation

Hypocomplementemic urticarial vasculitis syndrome is a rare condition, affecting mostly females (female–male ratio of 8:1) in the fourth decade [6].

The original diagnostic criteria proposed by Schwartz et al. (1982) are outlined in Table 2. Both major criteria, two minor criteria and exclusion of other systemic disease are requested for a positive diagnosis. However, as systemic manifestations, hypocomplementemia and anti-C1q antibodies are also seen in systemic disease, e.g., systemic lupus erythematosus (LES), Sjögren syndrome, and cryoglobulinemia, and these should be excluded for a positive diagnosis (Table 2) [1,4,12].

Chronic urticarial-like lesions are erythematous wheals, lasting longer than urticaria (usually 24–72 h), more frequently painful and burning than itchy, and can resolve with purpura and faint hyperpigmentation. Angioedema is frequently associated. Urticarial lesions are a major diagnostic criterion but, in some instances, could be preceded by systemic manifestations and hypocomplementemia. Biopsy of active skin lesions is the gold standard of diagnosis. Biopsy reveals small vessel vasculitis, associating leukocytoclasis (fragmentation of leukocytes with nuclear debris), endothelial swelling and vessel wall destruction with fibrinoid deposits, eventual red blood cell extravasation and perivascular granulocytic infiltration. C1q immunostaining is almost constant in active lesions [2,3,5,12,13].

Hypocomplementemia (low C1q, C3 and C4) is seen in almost all cases and is a marker of disease severity. However, C3 and C4 levels can fluctuate from undetectable to normally low during disease, despite a low C1q [14]. Anti-C1q antibodies were reported in all cases with active disease but have a low specificity, because are also seen in other autoimmune diseases. While antinuclear ANA are frequent (50–55%), anti-dsDNA excludes HUVS [3,12,13]. Urticaria with low complement levels could be seen in many systemic conditions, e.g., SLE, Sjögren syndrome, cryoglobulinemia. Accordingly, these conditions should be excluded for a HUVS diagnosis [6,12].

Common systemic manifestations are arthralgia/arthritis (70–80%) and ocular involvement (episcleritis, uveitis, conjunctivitis—10–56%), whereas respiratory (COPD, chronic-obstructive pulmonary disease), gastrointestinal (abdominal pain, vomiting, diarrhea) and renal involvement (hematuria, proteinuria, decline in kidney function) are less common (20%, 18–30% and 14–50%, respectively) [3,13].

More than half of patients had constitutional symptoms (fever, malaise, fatigue) and inflammation (high erythrocyte-sedimentation rate, ESR, and C-reactive protein, CRP) is common [12].

The outcome is generally good but seems negatively influenced by COPD and kidney involvement [6,13].

### 3.2. HUVS Pathogenesis

C1q is a glycoprotein composed of six identical subunits, each formed by three chains (A, B and C) assembled in a structure with a globular head and a collagen-like triple-helix tail [15,16]. The globular end of C1q functions as a pattern recognition molecule for immunoglobulin and polyanionic molecules, e.g., apoptotic bodies. Binding to the heavy chains constant region (Fc) of immunoglobulins IgG and IgM induces a conformational change in the C1q collagen-like stalk. To note, C1q binds only to immunoglobulins in immune complexes, as an immunoglobulin conformational change is necessary to expose its Fc. C1r then autoactivates and in turn activates C1s allowing C1q to associate with C1r and C1s to form C1, the first component on the classical pathway of complement activation (Figure 4) [16].

As activation of complement facilitates the clearance of immune complexes, the C1q main physiological function seems to be the clearance of immune complexes and apoptotic bodies. This is supported by data associating C1q deficiency, apoptotic bodies’ clearance and autoimmunity in SLE, both in experimental models and in humans [17,18,19,20,21].

Anti-C1q antibodies were initially detected in patients with SLE but thereafter were found in many autoimmune diseases, with a variable prevalence: up to 100% in HUVS, 30–60% in SLE, and 32% in rheumatoid vasculitis [15,22,23]. Also, they have been described in hepatitis C virus and HIV infections [23].

Anti-C1q antibodies bind to the collagen-like tail of C1q, not to its globular end, and there are particularities of the anti-C1q binding to the C1q depending on the underlying disease [16]. In Western Blot analyses, anti-C1q antibodies from patients with HUVS and overlapping syndromes reacted specifically with separated B and C protein chains, while those from patients with SLE did not, which underlines the pathogenetic differences between HUVS and SLE and may be in the future a useful tool to differentiate these two conditions [24].

Notably, the affinity of anti-C1q antibodies to C1q is very weak in plasma (liquid phase) but high when C1q is bound to immune complexes (solid phase), because binding to immunoglobulins changes the conformation of C1q, exposing a neoepitope (which is hidden in the fluid phase) for anti-C1q antibodies binding [25]. This could explain the normal levels of plasma C1q even when the titer of anti-C1q antibodies is high [22].

On the other hand, this suggests that anti-C1q antibodies may be pathogenic only when C1q molecules are bound in immune complexes deposited in tissues. This hypothesis was confirmed by Trouw et al. in a murine model [26]. Administration of a monoclonal antibody recognizing the collagen-like tail of the mouse-C1 (mAb JL-1) resulted in glomerular deposition and mild granulocyte influx, but without overt renal damage [26]. However, if mAb JL-1 was administered together with a sub-nephritogenic dose of C1q-fixing anti–glomerular basement membrane (anti-GBM) antibodies, the renal damage amplified, with major histological changes, important local inflammatory infiltrate and increased albuminuria [26]. Moreover, the renal damage was not observed in mice receiving a non–C1q-fixing anti-GBM preparation [26]. Accordingly, anti-C1q antibodies seem to act as an acquired amplification loop of the classical pathway activation, similar to C3-nephritic factor, the only molecule previously known to amplify the classical pathway [27].

Furthermore, prolonged exposure to the C1q epitope—unmasked by binding to immune complexes—could eventually generate auto anti-C1q antibodies [7].

Thus, anti-C1q are clearly involved in HUVS but what triggers their generation is unclear. A viral infection is a possibility, as anti-C1q antibodies were reported in viral disease (hepatitis C and HIV) [22].

Finally, activated complement generates vasculitis which causes organ lesions. The clinical expression is dictated by the vascular territory involved. Consequently, skin lesions could exist or not, and biopsy-proved vasculitis in any territory is diagnostic.

### 3.3. Renal Involvement in HUVS

The frequency of kidney involvement in HUVS is uncertain, partly because HUVS is a rare disease, partly because kidney evaluation, including biopsy, has not been consistently reported. In a series from Sweden, 50% of patients had hematuria or proteinuria, while 20% had abnormal urine examination in a previous review by Kobayashi et al., and a French survey reported renal involvement in 14% of cases [3,8,28]. Similarly, frequency of the type of glomerular lesions is difficult to evaluate, as kidney biopsy was reported in 33–67% of HUVS cases with abnormal urine examination [15,28].

We searched the literature from 1976 until 2020 for reported cases of HUVS with renal manifestations. We identified 92 cases and retained 60 for analyses, which had data available for individual patients (Table 3) [7,8,11,15,28,29,30,31,32,33,34,35,36,37,38,39,40,41,42,43,44,45,46,47,48,49,50,51,52,53,54,55,56,57,58,59,60,61,62,63,64,65,66,67,68,69].

In selecting patients, we did not use skin biopsy as an exclusion criterion, considering that HUVS is a systemic condition, in which the timetables of various organ lesions can differ. Supporting this, 4 out of 48 skin biopsies did not show leukocytoclastic vasculitis but, in one of these, vasculitis was seen at a gastrointestinal biopsy [33,52,68,69]. Moreover, in one case renal involvement preceded urticarial lesions, as in our first case, where skin lesions were not present, and in both these cases cellular crescents suggested vasculitis [11]. Additionally, in our second case, the cutaneous lesions were rather discreet and kidney biopsy showed vasculitis. Accordingly, urticarial lesions are clinically indicative for HUVS, while a biopsy showing features of leukocytoclastic, necrotizing vasculitis, irrespective of territory, in the presence of anti-C1q antibodies and hypocomplementemia is diagnostic, even in the absence of skin lesions, after excluding other autoimmune conditions.

HUVS was classically described as a condition affecting predominantly women in the fifth decade and only rarely children [1,12,13]. In the cohort of HUVS patients with renal involvement we evaluated (HUVS-KI), 18% were under the age of 18. In adults, the median age and proportion of women were lower than reported in unselected patients (median age 41 vs. 42–51 years and 65 vs. 88%), and the median age at diagnosis was lower in women than in men (40, 95%CI 35–44 vs. 52, 95%CI 36–63 years) [3,8]. Moreover, the women to men ratio was five times higher in adults than in pediatric patients (2.7 vs. 0.5; *p* = 0.01). Thus, kidney involvement seems more frequent in male pediatric patients and in adult women, which are affected at lower ages than men, suggesting a relation between age, sex and kidney involvement in HUVS.

Hematuria, proteinuria and altered kidney function in various combinations were reported in 5–20% of unselected patients with HUVS [28]. In our analysis of HUVS-KI patients, hematuria in association with proteinuria was the most frequent (70%) abnormality. Isolated hematuria and isolated proteinuria were uncommon (7 and 17%) (Table 4). There were no differences between adult and pediatric patients. Hence, urinalysis seems a good screening test for kidney involvement in HUVS.

Abnormal kidney function (GFR < 60 mL/min) was reported in one third of HUVS-KI cases but was available in less than half. Unfortunately, we were unable to identify in the reported cases the rapid progressive course seen in all our cases. However, the possibility of a rapidly progressive course should be borne in mind, as illustrated by the cases we presented. Accordingly, kidney function should also be monitored in HUVS patients, to avoid late nephrology referral and irreversible loss of kidney function (see Case 3).

Kidney biopsy was performed frequently, as data were available in 52 cases (Table 5). The most frequent glomerular pattern of injury was membranoproliferative (35%), followed by mesangioproliferative (21%) and membranous (19%) but was not specified in 42% of cases. Crescents were found in 23% of cases in which GFR was significantly lower (20 vs. 61.5 mL/min/1.73m2; *p* = 0.04). Thus, the membranoproliferative pattern seems the most frequent in HUVS-KI patients; crescents formation is not rare and is associated with more altered kidney function and was seen in two of our cases.

Immunofluorescence was available in 38 patients, with a full-house pattern, as in two of our cases. The association between full house immunostaining (staining for all immune reactants, IgG/IgA/IgM/C1q/C3) and membranoproliferative, mesangioproliferative or membranous pattern is highly suggestive for lupus nephritis, which explains why some authors consider HUVS as a SLE-related syndrome [13].

In terms of treatment, there is no consensus, as the available data is sparse. In a French retrospective survey, Azathioprine, Mycophenolate mofetil and Cyclophosphamide provided similar efficacy, while Rituximab was associated with a longer duration of remission without add-on therapy as compared to corticosteroids or conventional immunosuppressive therapy [3].

In patients with HUVS-KI, immunosuppressive regimens used most frequently (86% of cases) included corticosteroids in high doses, orally (74%) or intravenously (IV) (12%). The second most frequently used drug was Cyclophosphamide (34%, 12% IV), followed by Azathioprine (22%), Mycophenolate mofetil (15%) and Cyclosporine (8%). Occasionally, plasma exchange (8%) and IV immunoglobulin (7%) were prescribed. Notably, three patients developed serious infections (sepsis) related to immunosuppression and two of them died.

The heterogeneity of IS regimens suggests that therapy was guided by the clinical presentation: induction therapy with corticosteroids and Cyclophosphamide, eventually reinforced by plasma exchange or IV immunoglobulin, followed by corticosteroids and Azathioprine for maintenance in severe systemic vasculitis or rapidly progressive glomerulonephritis, Cyclosporine-based regimens for nephrotic syndrome and Mycophenolate based regimens for mesangioproliferative glomerulonephritis.

All our patients received monthly pulses of IV Cyclophosphamide and Methylprednisolone, followed by oral Prednisone, as in the induction therapy for severe systemic vasculitis. One patient received maintenance therapy with Rituximab (100 mg/ every 3 months) with partial remission of the nephrotic syndrome but with persistent immunological activity. Another patient was unresponsive to initial IS course and underwent plasma exchange followed by maintenance therapy (Azathioprine and low-dose steroids) and achieved complete renal response. Only one patient progressed to end-stage renal disease (ESRD). This patient was referred late to the nephrologist (two months after renal function declined) and the aggressivity of IS therapy was limited by the hematological side-effects. In our experience, the outcome can be improved by aggressive immunosuppressive treatment.

A significant association between skin lesion and immunological response was found in a retrospective survey [3]. Supporting this, in our unresponsive patient two flares of urticarial lesions appeared under IS.

Regarding kidney outcome, in our analysis of HUVS-KI patients progression to ESRD was reported in similar proportion in adults and children (15 and 17%; *p* = 0.9) and was in relation with kidney function at diagnosis (all patients progressing to ESRD had a GFR < 30 mL/min, while none of those with GFR ≥ 30 mL/min did so; *p* = 0.002).

Death occurred in 26% of adults and 8% of children (*p* = 0.2) and was due mainly to respiratory causes (severe respiratory failure, hemorrhagic alveolitis, respiratory sepsis). Severe kidney involvement seems to increase the risk of death, as 44% of deaths were reported in ESRD patients, while 18% were reported in patients without ESRD (*p* = 0.08). Kidney involvement appears to negatively influence the outcome, as in a cohort from Sweden ESRD and death were each reported only in 2% of HUVS cases [8]. Thus, as in other vasculitis, the outcome in HUVS is driven by the severity of organ lesion, in this case the kidney.

In conclusion, HUVS is a rare small vessel vasculitis mediated by anti-C1q antibodies, which can produce an immune-complex mediated glomerulonephritis, possibly with a rapidly progressive course. As in other systemic vasculitis, renal involvement implies a poorer prognosis, but the outcome can be improved by aggressive immunosuppressive treatment.

## Figures and Tables

**Figure 1 jcm-09-02131-f001:**
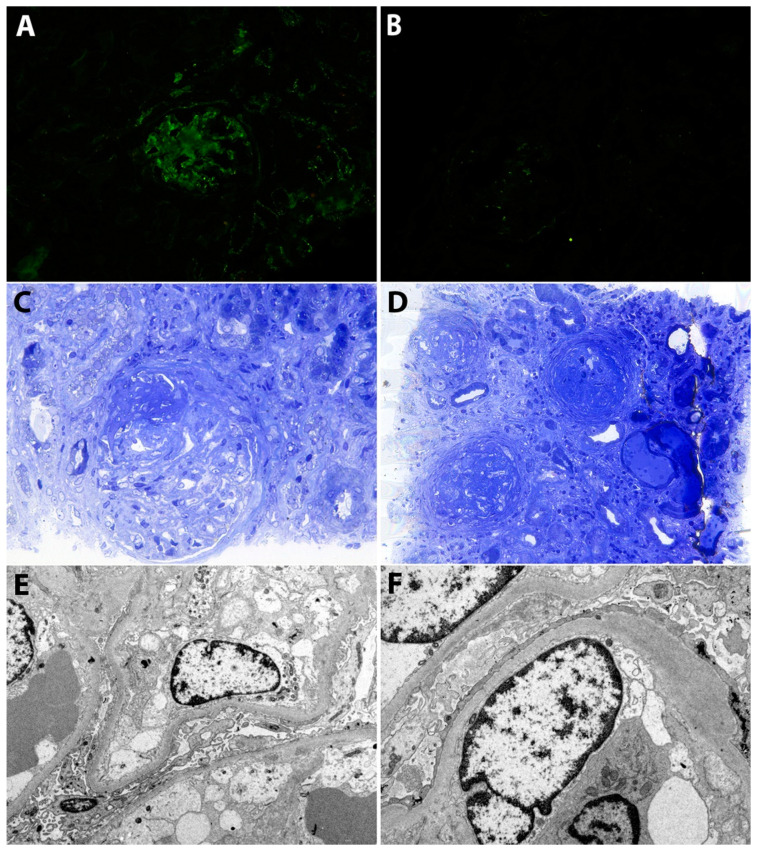
(**A**) Immunofluorescence. Moderate staining for IgG in the mesangium and along the glomerular capillary walls. (**B**) Immunofluorescence. Minimal staining for C3 in the mesangium and along the glomerular capillary walls. (**C**) Light microscopy (Toluidin Blue). Glomerulus with a membranoproliferative pattern of injury, fibrinoid necrosis and a fibro-cellular crescent. (**D**) Light microscopy (Toluidin Blue). Glomeruli showing a membranoproliferative pattern with diffuse lobular accentuation (mesangial expansion with hypercellularity, endocapillary hypercellularity and fibrous crescents). There is also moderate tubular atrophy and interstitial fibrosis. (**E**,**F**) Electron microscopy. There are subendothelial, unstructured, electron-dense deposits. Podocytes show diffuse foot process effacement. Additionally, there is severe endothelial injury with swollen glomerular endothelial cells and loss of fenestrations.

**Figure 2 jcm-09-02131-f002:**
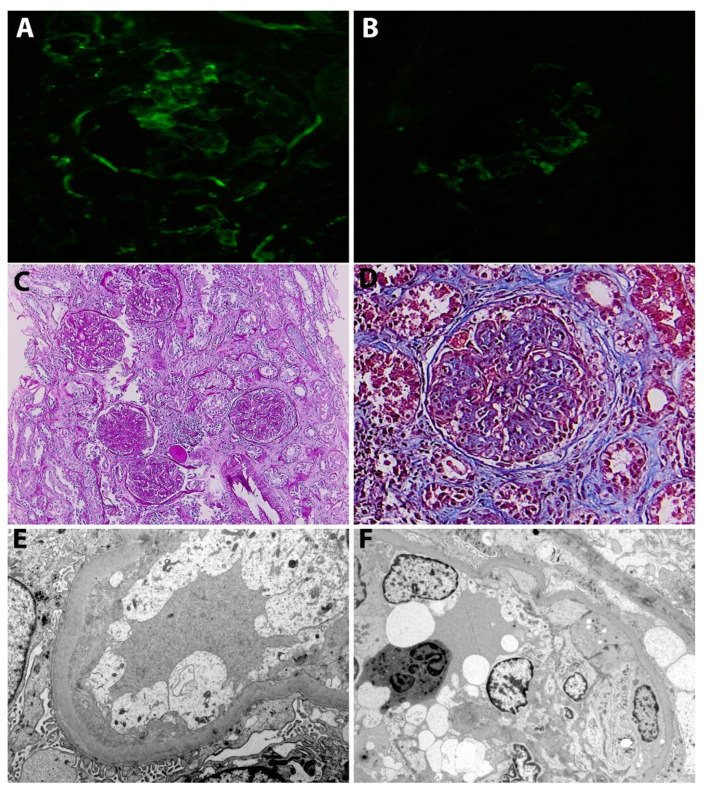
(**A**) Immunofluorescence. Moderate staining for IgG in the mesangium and along the glomerular capillary walls. (**B**) Immunofluorescence. Moderate staining for C3 in the mesangium and along the glomerular capillary walls. (**C**) Light microscopy (PAS staining). Glomeruli showing a membranoproliferative pattern with diffuse lobular accentuation (mesangial expansion with hypercellularity, endocapillary and extra-capillary hypercellularity). There is also moderate tubular atrophy and interstitial fibrosis and arteriosclerosis. (**D**) Light microscopy (Masson staining). Glomerulus with a membranoproliferative pattern of injury and a fibro-cellular crescent. Additionally, there is mesangial expansion due to accumulation of extracellular matrix, consistent with lesions of class II diabetic nephropathy (**E**) Electron microscopy. Podocytes showing diffuse foot process effacement. Additionally, there are swollen glomerular endothelial cells with loss of fenestrations and glomerular basement membrane thickening (460 nm). (**F**) Electron microscopy. Podocytes showing diffuse foot process effacement. There is also severe endothelial injury and endocapillary hypercellularity.

**Figure 3 jcm-09-02131-f003:**
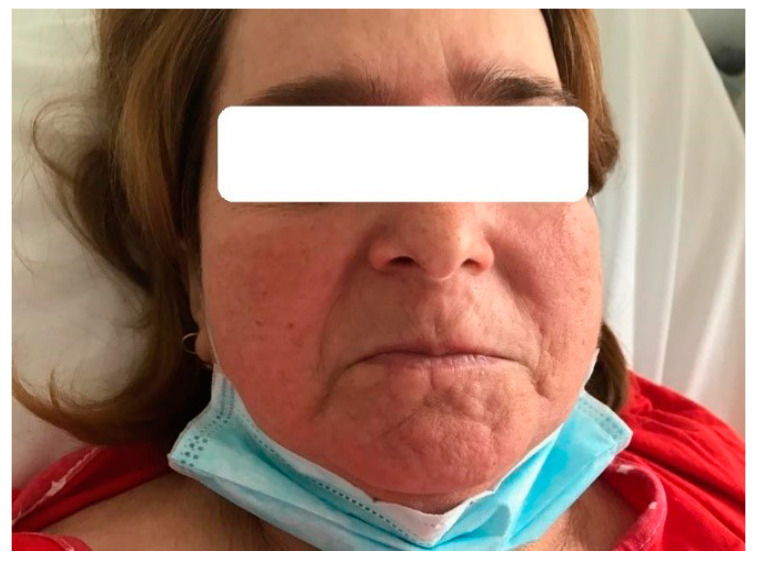
Urticarial-like exanthema (case 3).

**Figure 4 jcm-09-02131-f004:**
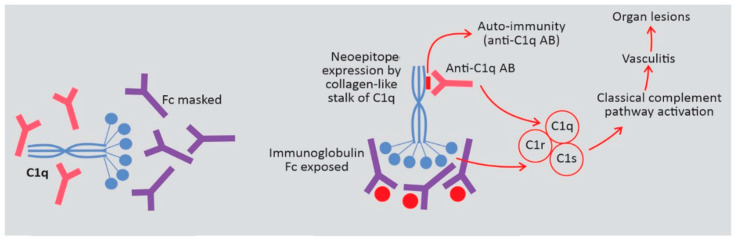
Possible pathogenetic roles of anti-C1q antibody (anti-C1q-AB) in hypocomplementemic urticarial vasculitis syndrome (HUVS).

**Table 1 jcm-09-02131-t001:** Laboratory data in reported cases.

	At Presentation			Follow-Up					
				6 mo	6 mo	6 mo	27 mo	16 mo	29 mo
Case	1	2	3	1	2	3	1	2	3
Serum creatinine [mg/dL]	4.29	1.27	5.96	1.16	0.86	3.9	0.9	0.85	3.9
eGFR (ml/min/1.73 m^2^)	12	50	8	56	81	13	76	82	13
Serum albumin [g/dL]	2.8	1.3	2.8	3.5	2.41	4	4.61	3.3	4
Proteinuria [g/24 h]	13.4	6.2	4	7.4	2	1.2	1.8	2.6	1.2
Hematuria [RBC/μL]	1937	105	773	49	39	25	10	14	25
Anti-C1q antibodies [U/mL]	75.8	356	112.8	3.5	72	75	9.1	157	86
C3 [mg/dL]	58.4	41	18.7	103	65.1	81.3	108	65.7	81.3
C4 [mg/dL]	9.24	8.94	3.61	31.9	19.6	23.3	30.2	17.1	23.3
CRP [mg/L]	37.5	5.06	12.9	1.74	0.43	8	2.15	0.37	8

Abbreviations: eGFR, estimated glomerular filtration rate; RBC, red blood cells; CRP, C-reactive protein; mo, months. Normal ranges: C3, 90–180 mg/dL; C4, 10–40 mg/dL; C-reactive protein, 0–3 mg/L.

**Table 2 jcm-09-02131-t002:** Diagnostic criteria for HUVS (modified from [1,12]).

**Major Criteria**
● Chronic urticarial exanthema
● Hypocomplementemia
**Minor criteria**
● Leuko-cytoclastic vasculitis
● Arthralgia/arthritis
● Uveitis/episcleritis/conjunctivitis
● Glomerulonephritis
● Abdominal pain
● Positive C1q antibodies
For a positive diagnosis are necessary:
Two major criteria, two minor criteria and exclusion of an autoimmune disease (SLE, Sjögren syndrome, cryoglobulinemia)

**Table 3 jcm-09-02131-t003:** Reported cases of HUVS for whom individual data were available.

	Author (Year)	Age	Sex	Organ Damage *	Anti-C1q AB	GFR	Renal Manifestations	Kidney Biopsy	Immunosuppressive Therapy	ESRD	Death
1	Sissons (1974) [29]	48	F	AE	NR	8	NN proteinuria	GN	CS (PO), CYC (PO)	No	No
2	Feig (1976) [30]	31	F	J, E, L	NR	NR	Hematuria, NN proteinuria	GN	NR	NR	NR
3	Ludivico (1979) [31]	24	F	D	NR	90	Hematuria, N Syndrome	MPGN	CS (PO, IV), AZA	No	No
4	Schultz (1981) [32]	36	M	J, E	NR	96	Hematuria, NN proteinuria	MesPGN	CS (PO)	No	No
5	Schultz (1981) [32]	54	F		NR	64	Hematuria, NN proteinuria	MesPGN	CS (PO)	No	No
6	Meyrier (1984) [33]	58	F	J, D, AE	NR	NR	NS proteinuria	GN	CS	No	No
7	Waldo (1985) [34]	16	M	D	NR	NR	Hematuria, NS proteinuria	MPGN	CS (PO)	Yes	No
8	Kobayashi (1985) [28]	55	M	E	NR	96	Hematuria, N Syndrome	MN	CS (PO)	No	No
9	Kobayashi (1985) [28]	28	F		NR	102	Hematuria, NS proteinuria	MN	CS (PO)	NR	NR
10	Fortson (1986) [35]	45	F	J, L	NR	68	Hematuria, NS proteinuria	MPGN	CS (PO)	No	No
11	Ramirez (1987) [36]	59	M	E	+	20	Hematuria, NN proteinuria	MPGN	CS (PO), CYC, AZA	No	Yes
12	Wisnieski (1994) [37]	32	F	J, E, L	+	85	NN proteinuria	NR	CS (PO), HCQ, IG	No	No
13	Wisnieski (1994) [37]	31	F	J, E, AE	+	25	N Syndrome	GN	CS (PO), HCQ, IG	No	No
14	Martini (1994) [38]	14	M	J, E, D	+	NR	Hematuria, NS proteinuria	MesPGN, (FC)	CS (PO, IV), CYC (PO, IV)	Yes	No
15	Martini (1994) [38]	15	M	J, E, L, D, AE	+	NR	Hematuria	NR	CS (PO, IV), CYC (IV), AZA	No	No
16	Mituiki (1994) [39]	66	M	L	+	44	N Syndrome	MN	CS	No	Yes
17	Wisnieski (1995) [15]	53	M	J, E, L, AE	+	107	Hematuria, NN proteinuria	MPGN	CS (PO, IV)	No	Yes
18	Wisnieski (1995) [15]	41	F	J, L, AE	+	92	NN proteinuria	MesPGN	NR	No	Yes
19	Wisnieski (1995) [15]	40	F	J, L, AE	+	37	Hematuria, NS proteinuria	MPGN	AZA	No	Yes
20	Wisnieski (1995) [15]	37	F	J, L	+	NR	Hematuria, NN proteinuria	NR	CS (PO), CYC	No	Yes
21	Wisnieski (1995) [15]	26	F	J, E, D, AE	+	69	Hematuria	MesPGN	CS (PO), HCQ	No	No
22	Wisnieski (1995) [15]	35	F	J, L, AE	+	95	Hematuria, N proteinuria	MN	MET	No	Yes
23	Eiser (1997) [40]	35	F	J, E, L	NR	NR	Hematuria, NN proteinuria	MN	CS (PO), CYC (IV)	No	No
24	Renard (1998) [41]	13	M	J, E	NR	NR	Hematuria, N proteinuria	MesPGN (FC, CC)	CS (PO, IV), CSA, AZA	No	No
25	Jovanovic (1999) [42]	41	F	J	NR	NR	Hematuria, NS proteinuria	MesPGN	CS (PO)	No	No
26	Trendelenburg (1999) [7]	37	F	J, E, D, L	+	NR	Hematuria, NS proteinuria	MesPGN	CS (PO), HCQ, AZA, PLEX, IVIG	No	No
27	Soma (1999) [43]	43	M	NR	NR	109	Hematuria, N Syndrome	MN	CS (PO, IV), CSA	No	No
28	Cadnapaphornchai (2000) [44]	11	F	J	NR	NR	Hematuria, NN proteinuria	MPGN	CS (PO)	No	No
29	Sessler (2000) [45]	40	F	J, AE	+	NR	NR	MPGN	CS, CYC	No	No
30	Messiaen (2000) [46]	27	F	J, E, L	NR	9	Hematuria, NS proteinuria	MPGN (FC)	CS, CYC	Yes	No
31	Chew (2000) [47]	55	F	L, AE	+	90	NN proteinuria	MPGN	CS, CYC (IV), MET	No	No
32	Boulay (2000) [48]	34	F	L	NR	22	NR	MPGN	CS	No	Yes
33	El Maghraoui (2001) [49]	41	F	J	NR	NR	NN proteinuria	MN	CS (PO), HCQ, CYC (PO)	No	No
34	Brass (2001) [50]	44	F	J, E, AE	+	NR	Hematuria, NN proteinuria	MN	CS (PO), HCQ, CYC, CSA, MET, IVIG	No	No
35	Saeki (2001) [51]	49	M		NR	59	Hematuria, N Syndrome	MPGN	CS (PO)	No	No
36	Grimbert (2001) [52]	36	M	J, AE	+	15	Hematuria, N proteinuria	MPGN	CS (PO, IV), CSA, AZA, PLEX	Yes	Yes
37	Toprak (2004) [53]	53	F	D	NR	9	Hematuria, NN proteinuria	GN	CS (PO, IV), CYC (IV)	Yes	Yes
38	Enriquez (2005) [11]	39	F	J, E	+	71	Hematuria, N Syndrome	MPGN (C)	CS, CYC, MMF	No	No
39	Wiederkehr (2006) [54]	63	M	J, E, L, AE	NR	42	Hematuria, N Syndrome	MN	CS (PO), MMF	No	No
40	Balsam (2008) [55]	23	F	J, D	+	29	Hematuria, N Syndrome	GN (C)	CS (PO, IV), CYC (IV), PLEX	Yes	No
41	Özçakar (2010) [56]	6	F	D	NR	NR	Hematuria, NN proteinuria	GN (C)	CS, CYC, AZA	No	Yes
42	Al Mosawi (2013) [57]	8	M	J, E, AE	NR	NR	Hematuria, NN proteinuria	GN (C)	CS (PO, IV), MMF, AZA	No	No
43	Loricera (2014) [2]	35	F	J	NR	NR	Hematuria	NR	CS (PO), HCQ	No	No
44	Park (2014) [58]	30	M	J, AE	NR	44	Hematuria, NN proteinuria	MPGN	CS (PO, IV), HCQ, CYC (PO), MMF	No	No
45	Pasini (2014) [59]	9	F	J, E, D	+	NR	Hematuria	MesPGN	CS (PO), MMF	No	No
46	Pasini (2014) [59]	9	F	J, E, L, D	+	NR	Hematuria, NN proteinuria	GN	CS (PO), CYC (PO), AZA	No	No
47	Pasini (2014) [59]	9	M	J, E, D, L, AE	+	NR	Hematuria, N Syndrome	GN (C)	CS (PO, IV), HCQ, CYC (IV), CSA, AZA	No	No
48	Zakharova (2016) [60]	32	F	E, D	+	24	Hematuria, NN proteinuria	MesPGN (FC)	CS (PO, IV), AZA	No	No
49	Jung (2017) [61]	15	M	E	NR	NR	NN proteinuria	MN	CS (PO, IV), HCQ, AZA, TAC	No	No
50	Gheerbrant (2017) [62]	41	F	E, D	+	69	Hematuria, N Syndrome	GN	CS (PO, IV), MMF	No	No
51	Tanaka (2017) [63]	64	F	D	NR	22	Hematuria, NN proteinuria	GN (CC)	CS (PO, IV)	No	No
52	AlHermi (2017) [64]	6	M	J, E, D	+	NR	Hematuria, NS proteinuria	MesPGN	CS (PO, IV), AZA	No	No
53	Salim (2018) [65]	31	F	J, D	+	12	Hematuria, N Syndrome	MPGN (C)	CS (PO, IV), CYC (IV), MMF, R	No	No
54	Hopkins (2018) [66]	66	M	J, D	NR	5	Hematuria, NS proteinuria	NR	CS (PO)	No	No
55	Sjowall (2018) [8]	53	F	L	+	NR	NR	NR	CS (PO), HCQ, MMF	Yes	Yes
56	Sjowall (2018) [8]	64	F	J	+	NR	NR	NR	CS (PO), CYC, PLEX	Yes	Yes
57	Sjowall (2018) [8]	49	F	J	+	NR	NR	NR	CS (PO), HCQ, MMF	No	No
58	Lopez-Romero (2019) [67]	52	M	J, E	NR	31	Hematuria, N proteinuria	MPGN	CS (PO, IV)	No	No
59	Ueki (2020) [68]	36	M	D, L	+	22	Hematuria, N proteinuria	MPGN (CC)	CS (PO, IV), PLEX	Yes	No
60	Boyer (2020) [69]	49	F	J	+	35	Hematuria, N proteinuria	FSGS	CS (PO), HCQ	No	No

* Except kidney and skin damage which were seen in all cases; C—crescents not specified; CC—cellular crescents; FC—fibrous crescents; GFR—Glomerular Filtration Rate [mL/min/1.73m^2^]; ESRD—End Stage Renal Disease; F—female; M—male; AE—angioedema; J—joints; E—eye; L—lung; D—digestive tract; NN proteinuria—non-nephrotic proteinuria; N proteinuria—nephrotic proteinuria; NS proteinuria—non-specified proteinuria; N syndrome - nephrotic syndrome; GN—non-specified glomerulonephritis; MPGN—membrano-proliferative glomerulonephritis; MesPGN—mesangio-proliferative glomerulonephritis; MN—membranous nephropathy; FSGS—focal-segmental glomerulosclerosis; NR –not reported; CS—Corticosteroids; PO—per os; IV—intravenous; CYC—Cyclophosphamide; AZA—Azathioprine; HCQ—Hydroxychloroquine; IVIG—Immunoglobulins, MET—Methotrexate; CSA—Cyclosporine A; PLEX—plasma exchange; MMF—Mycophenolate mofetil; TAC—Tacrolimus; R = Rituximab.

**Table 4 jcm-09-02131-t004:** Renal abnormalities in HUVS patients with kidney involvement (*n* = 60).

Hematuria	76.7% (46)	Isolated 7% (4)		
		Associated 70% (42)	Proteinuria (Non-Nephrotic)	27% (16)
			Proteinuria (Nephrotic)	10% (6)
			Nephrotic Syndrome	17% (10)
			Non-Specified Proteinuria	17%(10)
Proteinuria (isolated)	12% (7)			
Nephrotic syndrome	3% (2)			
Not reported	8% (5)			
GFR	Mean; 95% CI	44; 25–69		
	<60	33% (20)		
	Not reported	41% (25)		
Data in brackets are numbers of patients; GFR—Glomerular Filtration Rate [mL/min/1.73 m^2^];				

**Table 5 jcm-09-02131-t005:** Patterns of kidney lesions HUVS-KI (*n* = 52).

Membranoproliferative GN	35% (18)
GN not specified	23% (12)
Mesangioproliferative GN	21% (11)
Membranous nephropathy	19% (10)
Focal segmental glomerulosclerosis	2% (1)
Crescents	23% (12)

Data in brackets = number of patients; GN—glomerulonephritis.

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
