# Peer review of "Kidney Involvement in Hypocomplementemic Urticarial Vasculitis Syndrome—A Case-Based Review"

_jcm, 2020, doi:10.3390/jcm9072131_

Round 1

Reviewer 1 Report

The manuscript by  Ion  et al. describes three cases of hypocomplementemic urticaria vasculitis syndrome (HUVS) with clinically significant renal involvement. The authors also did a literature search to retrieve published cases of HUVS with kidney disease and performed a careful comparative analysis. The cases are well presented and the discussion is well organized and logical. However, several points must be clearly specified to fully appreciate the quality of the manuscript

Ultrasound kidney examination is not reported for any of the three patients. This is an important diagnostic step in the evaluation of kidney involvement. Did patients undergo kidney ultrasound evaluation ? All patients had normal finding ? How do the authors eventually explain this, given the significant histopathological alterations found in the biopsies.

Information on GFR for the three patients in Table 1 at baseline and follow up should be provided to compare them with patients in previously published papers.

The discussion on treatment strategies is somehow confusing. The authors should be much more precise in differentiating induction from maintenance treatment, otherwise  the message can be misleading. I understand that retrieval of information from previous (and sometime really old) papers may be difficult but nonetheless the drugs used for induction and treatments should be clearly differentiated. This should be specifically done also in Table 3.

Figure 4 requires a better explanation. The authors should clearly describe what do they mean by solid phase, e.g., is it endothelial surface ? or else ?

The text requires extensive English language revision.

Minor

Normal value of CRP should be indicated

Figure 4: “Lichid phase” is probably a mistake; “epitop” should be “epitope”

Line 349-350: please explain what does “full house” mean; for general readers this is an unusual term.

Line 389: ESDR should be ESRD.

Author Response

We are submitting the reply to the composition comments you have made on our manuscript entitled “Kidney involvement in hypocomplementemic urticarial vasculitis syndrome – a case-based review?” coauthored by Oana Ion, Bogdan Obrișcă, Gener Ismail, Bogdan Sorohan, Sonia Bălănică, Gabriel Mircescu and Ioanel Sinescu.

We have revised the manuscript based on the comments made by the reviewers.

Together with revised manuscript here is our answer to the reviewer’s comments.

Reviewer 1

The manuscript by Ion et al. describes three cases of hypocomplementemic urticaria vasculitis syndrome (HUVS) with clinically significant renal involvement. The authors also did a literature search to retrieve published cases of HUVS with kidney disease and performed a careful comparative analysis. The cases are well presented and the discussion is well organized and logical. However, several points must be clearly specified to fully appreciate the quality of the manuscript

Thank for your comments and suggestions. We have revised the manuscript based on your recommendations. Together with revised manuscript here is our response.

Point 1. Ultrasound kidney examination is not reported for any of the three patients. This is an important diagnostic step in the evaluation of kidney involvement. Did patients undergo kidney ultrasound evaluation ? All patients had normal finding ? How do the authors eventually explain this, given the significant histopathological alterations found in the biopsies.

Response 1. All patient underwent ultrasound evaluation and the kidneys aspect was within normal limits. We have added the information at each patient. The severity of kidney lesions seen by light microscopy examination does not correlate with the macroscopic aspect of the kidneys. Only sometimes, in the setting of severe proliferative lesions (extracapillarry cellularity) the kidney may appear enlarged due to coexistant interstitial inflammation and edema or in the context of severe nephrotic syndrome due to interstitial edema, while patients with advanced CKD may show shrunken kidneys due to interstitial scarring. But most often, the macroscopic aspect of the kidneys remains normal even in the context of severe glomerular lesions

Point 2. Information on GFR for the three patients in Table 1 at baseline and follow up should be provided to compare them with patients in previously published papers.

Response 2. We have added the GFR values in Table 1.

Point 3. The discussion on treatment strategies is somehow confusing. The authors should be much more precise in differentiating induction from maintenance treatment, otherwise  the message can be misleading. I understand that retrieval of information from previous (and sometime really old) papers may be difficult but nonetheless the drugs used for induction and treatments should be clearly differentiated. This should be specifically done also in Table 3.

Response 3. Thank you for your comment. Indeed treatment strategies are somewhat confusing especially given the rarity of this disorder and the lack of adequate treatment guidelines. As such, the majority of case reports have inadequate reports of type, regimen, dosages of IS therapy. Given this finding, it is very difficult to systemize these treatments options since most authors have conducted the IS therapy based on their personal clinical experience. However, we tried to synthesize the therapeutic approaches in the discussion section in the following paragraph.

“The heterogeneity of IS regimens suggests that therapy was guided by the clinical presentation: induction therapy with corticosteroids and Cyclophosphamide, eventually reinforced by plasma exchange or IV immunoglobulin, followed by corticosteroids and Azathioprine for maintenance in severe systemic vasculitis or rapidly progressive glomerulonephritis, Cyclosporine-based regimens for nephrotic syndrome and Mycophenolate based regimens for mesangioproliferative glomerulonephritis”.

In accordance, our center approach in the treatment of these disorders is similar to treatment guidelines of ANCA-associated vasculitis or lupus nephritis, and we clearly delineated in our patients the timeline of treatment approach.

Point 4. Figure 4 requires a better explanation. The authors should clearly describe what do they mean by solid phase, e.g., is it endothelial surface ? or else ?

Response 4. We agree that the description of solid and liquid phases are somewhat confusing. They were introduced initially with the intention to separate the plasma compartment and tissue compartment However, in order to avoid any confusion, we eliminated these descriptions from the figure.

Point 5. The text requires extensive English language revision.

Response 5. Thank you. We have corrected the grammatical errors throughout the text.

Minor

Point 6. Normal value of CRP should be indicated

Response 6.  We have added the normal CRP range

Point 7. Figure 4: “Lichid phase” is probably a mistake; “epitop” should be “epitope”

Response 7. We have corrected the errors. In addition, we have eliminated the term liquid and solid phase as it may give confusion (initially intended to signify the plasma compartment and tissue compartment).

Point 8. Line 349-350: please explain what does “full house” mean; for general readers this is an unusual term.

Response 8. We have detailed what full house means

Point 9. Line 389: ESDR should be ESRD

Response 9. We have made the correction

We hope that we have addressed all the issues of your comments.

Sincerely Yours,

Gener Ismail MD, PhD

Corresponding author: Assoc. Prof. Gener Ismail, MD, PhD – Department of Nephrology, Fundeni Clinical Institute, 258 Fundeni Street, District 2, Bucharest, Romania, zip code 022328; [email protected]

Reviewer 2 Report

This a well written, if somewhat long, cases series of 3 patients with HUVS and kidney involvement, and a review of the literature of this rare presentation. The authors do a good job of synthesizing the existing literature.

Major points

  1. Case 1 – there is a major question mark over whether this represents a case of HUVS, most notable the patient had no urticaria. The authors discuss that kidney disease may precede the urticarial lesions. The case satisfies other components (hypocomplementemia, MPGN pattern on biopsy, anti-C1q antibodies) but there is no gold standard test for this condition. The biopsy essentially shows an immune complex MPGN for which there is a broad differential diagnosis (which the authors tried to address), but we see patients like this with IC-MPGN where we cannot identify the underlying etiology, and specifically similar cases with lupus-like pathology who are ANA negative. Notably anit-C1q is not a specific test for HUVS and is commonly found in other autoimmune disorders. This patient would not fuflill the Scwhartz criteria used in Case 2 and 3

Minor points

  1. Case 1
    1. Diuresis of 1000mls/day, omit
    2. Patient had inflammatory syndrome (need to be more specific)
    3. Paraprotein workup – details of this need to be described
    4. Path – was there GBM duplication? Was there any vascular abnormalities on biopsy. Was there any substructure to the electron dense deposits on EM.
  2. Figure 1 & 2: Not sure showing the biopsy images for two patients adds useful information from a single patient. Much of the legend is duplicated. Only show path that highlights a key difference.
  3. Figure 4: spelling (liquid phase), neoepitope,
  4. Discussion – line 315: the statement “biopsy showing anti-C1q vasculitis is diagnostic, irrespective of territory” This needs re-written. The presence of necrosis / crescents with C1q deposition is not an anti-C1q vasculitis, and this statement is misleading.
  5. Discussion – overall this is too long. Would shorten the pathogenesis section,

Author Response

We are submitting the reply to the composition comments you have made on our manuscript entitled “Kidney involvement in hypocomplementemic urticarial vasculitis syndrome – a case-based review?” coauthored by Oana Ion, Bogdan Obrișcă, Gener Ismail, Bogdan Sorohan, Sonia Bălănică, Gabriel Mircescu and Ioanel Sinescu.

We have revised the manuscript based on the comments made by the reviewers.

Together with revised manuscript here is our answer to the reviewer’s comments.

Reviewer 2

Thank for your comments and suggestions. We have revised the manuscript based on your recommendations. Together with revised manuscript here is our response.

Major points

Point 1. Case 1 – there is a major question mark over whether this represents a case of HUVS, most notable the patient had no urticaria. The authors discuss that kidney disease may precede the urticarial lesions. The case satisfies other components (hypocomplementemia, MPGN pattern on biopsy, anti-C1q antibodies) but there is no gold standard test for this condition. The biopsy essentially shows an immune complex MPGN for which there is a broad differential diagnosis (which the authors tried to address), but we see patients like this with IC-MPGN where we cannot identify the underlying etiology, and specifically similar cases with lupus-like pathology who are ANA negative. Notably anit-C1q is not a specific test for HUVS and is commonly found in other autoimmune disorders. This patient would not fuflill the Scwhartz criteria used in Case 2 and 3.

Response 1. Thank for the observation. However, like we specified in the case presentation and discussion section, there are case reports of anti-C1q vasculitis that do not fulfill all diagnostic criteria provided by Schwartz. In our case, the extensive work-up done for case 1 excluded any other diagnosis, as our clinical judgement considered that this case is a form of anti-C1q vasculitis given the association with high anti-C1q with hypocomplementemia, the membranoproliferative pattern of glomerular injury with vasculitis features (fibrinoid necrosis and extracapillary hypercellularity) and the absence of other features suggesting a different etiology. We do agree that there are cases of lupus-like nephritis with negative serology or other forms of immune complex MPGN without a definitive etiological diagnosis with an extensive clinical work-up, but this patient did not show at any time during the follow-up lasting several years any other clinical/laboratory features suggestive of a lupus nephritis or other etiology. We do believe that given the rarity of this disorder, there is a possibility of underdiagnosing of this disease if considering only those patients that strictly fulfill these criteria. Additionally, these criteria, proposed almost 40 years ago, have not been validated in dedicated cohorts of patients, again probably because of the rarity of the disorder. However, with increasing understanding of this disease pathogenesis we strongly feel that patients highly suggestive of having anti-C1q vasculitis to be considered as such, even though they do not fulfill all diagnostic criteria. This may suggest that a call for a revision of diagnostic criteria be made in order to fully capture the entire spectrum of this disorder (similar to revisions made to other type of autoimmune disorders).

Minor points

Point 1. Case 1

Diuresis of 1000mls/day, omit

Patient had inflammatory syndrome (need to be more specific)

Paraprotein workup – details of this need to be described

Path – was there GBM duplication? Was there any vascular abnormalities on biopsy. Was there any substructure to the electron dense deposits on EM.

Response 1. We have added the details. Regarding the biopsy findings the electron dense deposits were unstructured. We did not find any GBM duplication (due to the recent onset of the disease most probably). However, signs of arteriosclerosis were found in the second patient and we updated the biopsy description

Point 2. Figure 1 & 2: Not sure showing the biopsy images for two patients adds useful information from a single patient. Much of the legend is duplicated. Only show path that highlights a key difference.

Response 2. Thank you for your comment. However, there are several reasons for why we want to show the biopsy features of both patients. First of all, the second patient had the anti-C1q vasculitis superimposed on a class II diabetic nephropathy and we wanted to outline the occurrence of non-diabetic lesions in patients with diabetes mellitus with or without diabetic nephropathy. Second, we had different staining available for light microscopy in these two patients which may provide additional information regarding types of histological lesions.

Point 3. Figure 4: spelling (liquid phase), neoepitope,

Response 3. We have corrected the errors. In addition, we have eliminated the term liquid and solid phase as it may give confusion (initially intended to signify the plasma compartment and tissue compartment.

Point 4. Discussion – line 315: the statement “biopsy showing anti-C1q vasculitis is diagnostic, irrespective of territory” This needs re-written. The presence of necrosis / crescents with C1q deposition is not an anti-C1q vasculitis, and this statement is misleading.

Response 4. Thank you for the observation. Indeed, the statement is misleading, we rephrased this sentence.

Point 5. Discussion – overall this is too long. Would shorten the pathogenesis section

Response 5. We did some modifications on the discussion section. However, as this article is intended to be a comprehensive updated review on the pathogenesis and renal involvement in HUVS, we constructed the discussion section as a review based on our clinical experience as reported with these 3 cases. Additionally, when drafting the initial manuscript of this article we started with already a longer discussion section and finally kept only the information that we consider essential for this update of the disease.

We hope that we have addressed all the issues of your comments.

Sincerely Yours,

Gener Ismail MD, PhD

Corresponding author: Assoc. Prof. Gener Ismail, MD, PhD – Department of Nephrology, Fundeni Clinical Institute, 258 Fundeni Street, District 2, Bucharest, Romania, zip code 022328; [email protected]

Round 2

Reviewer 1 Report

The authors adequately addressed all criticisms.